# Understanding the Contribution of Community Organisations to Healthy Ageing and Integrated Place-Based Care: Evidence from Integrated Care Data

**DOI:** 10.3390/healthcare11212827

**Published:** 2023-10-26

**Authors:** Chris Dayson, Chris Damm, Jan Gilbertson, David Leather, Will Ridge

**Affiliations:** 1Centre for Regional Economic and Social Research, Sheffield Hallam University, City Campus, Sheffield S1 1WB, UK; c.damm@shu.ac.uk (C.D.); j.m.gilbertson@shu.ac.uk (J.G.); d.leather@shu.ac.uk (D.L.); 2Advanced Wellbeing Research Centre, Sheffield Hallam University, Health Innovation Campus, Sheffield S9 3TU, UK; 3Leeds Office of Data Analytics, Leeds Health and Care Partnership, NHS West Yorkshire Integrated Care Board, Wakefield WF1 1LT, UK; will.ridge@nhs.net

**Keywords:** community organisations, healthy ageing, frailty, health and social care integration

## Abstract

*(1) Background*. There is interest in the role community organisations can play to support healthy ageing and the integration of health and social care. This study explored the contribution community organisations can make to this goal through the Leeds (UK) Neighbourhood Networks (LNNs), a novel example of community-based support. *(2) Methods.* An observational study of 148 LNN beneficiaries compared to the Leeds population aged 64 and over (n = 143,418) using the Leeds Data Model, and an analytical resource developed to support care planning. Measures included demographic characteristics, Electronic Frailty Index (EFI), the number of long-term health conditions (LTCs), and public health management cohort categorisation. *(3) Results.* LNN’s are primarily focussed on older people who are fit (44 percent) or experiencing the onset of LTCs (27 percent) and/or mild frailty (41 percent). However, they also support smaller numbers of people with moderate/severe frailty (15 percent) and five or more long-term conditions (19 percent). *(4) Conclusions.* Community organisations are well placed to support the ambitions of integrated care by providing support for older people with mild to moderate health and care needs. They also have the capacity to support older people with more severe needs if resourced to do so.

## 1. Introduction

The global population has aged significantly in the past 70 years [1]. Life expectancy at birth has increased from 47 years in the middle of the twentieth century to around 71 years today. This figure is expected to rise further to around 78 years by the middle of the twenty-first century. In real terms, the global population aged 60 years and over has increased from 200 million (8 percent) to almost 1 billion (12 percent) in the past 70 years. By 2050, this figure is expected to increase further to around 2.1 billion people (21%). This rapid population growth has major implications for how health and social care is provided with concerns about the pressures it will place on healthcare systems that were not designed support such high numbers of people experiencing age-related illnesses and disease. In response, governments around the world are increasingly focussed on how healthcare systems can be reformed in order to promote ‘healthy ageing’ alongside the treatment of disease.

The World Health Organisation (WHO) defines healthy ageing as ‘the process of developing and maintaining the functional ability that enables wellbeing in older age’ [2]. Functional ability is the capabilities that enable people to be and do what they have reason to value, including their ability to meet their basic needs; to learn, grow and make decisions; to be mobile; to build and maintain relationships; and to contribute to society. These attributes are linked to intrinsic capacity (i.e., physical health), which, although likely to deteriorate with age, can be moderated by adapting the environment, including the communities in which older people live and participate, to reduce barriers and mitigate declining capacity.

Given the importance of neighbourhood and environmental factors in supporting healthy ageing, there is global interest in the role that community organisations can play in supporting older people to age in place [3,4,5] as part of new models of ‘age friendly’ integrated care [6,7,8]. However, the evidence base on this topic requires development if health and care systems are to make significant long-term investments in community work. Where there is good evidence, it tends to be qualitative in nature, and there is a lack of quantitative, controlled or comparative studies [9,10]. This article aims to begin addressing this gap in the context of the UK where the integration of placed-based primary and secondary healthcare (which is the remit of the universal National Health Service (NHS)) with social care (which is the preserve of local government) is a strategic policy goal [11,12].

The aim of health and social care integration is to join up care for people that has historically been delivered through fragmented services, improving the quality of care that people receive and the satisfaction of staff providing it [13]. Two specific aims of health and social care integration are of relevance to this article. First, it promotes place-based community-centred care to help people with disabilities, including those who are suffering from dementia and other mental health issues, to live independent and healthy lives. Second, it places an emphasis on unlocking the power of data across local authorities and the NHS to support the development of new and innovative services to tackle the problems facing communities. It is argued that a more joined up approach to data will bring NHS and social care services much closer together to maximise the opportunities for health improvement [14].

This article presents an observational comparative exploration of the work of the Leeds Neighbourhood Networks (LNNs) using a novel health and care dataset (the Leeds Data Model (LDM)) that has been developed to support health and social care integration [15]. The LNNs are a group of community organisations that have been established for more than 30 years. Each LNN is an independent community organisation that aims to support older people (referred to as their ‘members’) to live independently and to participate in their community through activities and services provided at a neighbourhood level [16]. There are 37 LNNs covering the whole of Leeds, a northern city in England with a socially, economically and ethnically diverse population of approximately 800,000 people. LNNs come in different shapes and sizes: some are small local community groups; others are medium-sized voluntary organisations who run the NN alongside a wider range of community-based activities and services; and two are large national organisations, including a housing and care home provider and a national older person’s charity, who run NNs as a complement to their core activities [17]. 

Although the form, function, activities and services of LNNs vary, they also share some key characteristics. First, they are all run with the involvement of older people: each LNN has a management committee with an emphasis on community representation. Second, their services typically combine social opportunities and physical activities with more targeted support (information and advice, advocacy, dementia support and frailty reduction) to improve or preserve health and wellbeing (see Table 1). For most members long-term support is provided, from periods ranging from 1 year to more than 10 years in some instances. The LNNs receive a core grant from Leeds City Council and the local NHS to address four major policy requirements associated with integrated place-based care: to reduce social isolation and loneliness; to increase contribution and involvement; to increase choice and control; and to enhance health and wellbeing. 

This study aimed to understand the contribution of community organisations to healthy ageing and integrated place-based care. Using the LNNs as a case study, and drawing on novel integrated-care data, the article answers the following questions:What are the demographic characteristics of LNN members and how do they compare to the wider Leeds population aged 64 and over?What are the health characteristics of LNN members according to key population health measures and how do they compare to the wider Leeds population aged 65 and over?What do these data tells us about the potential role of community organisations to address healthy ageing as part of integrated place-based care?

Research questions 1 and 2 are addressed in Section 3 (Results). Research question 3 is considered in Section 4 (Discussion). The research was undertaken as part of a wider long-term mixed-methods study of the Leeds Neighbourhood Networks and their contribution to healthy ageing and integrated place-based care [16,17,18].

## 2. Methods

We undertook an observational study of one LNN to test the use and applicability of the LDM for the undertaking of a comprehensive analysis of the implementation of integrated place-based care. The LDM is an analytical resource used across the Leeds Health and Care system to support Population Health Management and healthcare system planning. The data are pseudonymised so individuals cannot be re-identified, and it includes health and social care data from services including hospitals, GP practices, urgent care, ambulance, community and mental health, maternity, adult social care and population data. 

There is interest in how the LDM (and similar datasets in other areas of the UK) can be used to provide insights into the work that non-NHS and non-local authority providers, such as community organisations, undertake in support of integrated place-based care. Currently, however, the LDM does not include a mechanism for routinely identifying who has (and who has not) received support from these independent providers. As such, a process was developed with the LDM team to sample and pseudonymise LNN members within the data. To ease the resource burden and capture key learning, one LNN was selected in 2019 to take part in this study.

### 2.1. Stage 1: Sampling

A purposive sample of LNN members (n = 148) was selected over a 6 month period (June–December 2019). The sample was designed to be representative of LNN members in terms of age, gender, ethnicity and services accessed (see Section 3.1 for an overview). A comparative sample containing the full Leeds population aged 64 and over (n = 143,418) was obtained from the LDM.

### 2.2. Stage 2: Collecting NHS Numbers and Consent

Each UK citizen is allocated a personal NHS identification number (‘NHS number’) for their engagement with the healthcare system. NHS numbers enable data about each individual’s engagement with different services to be linked across the health and care system. NHS numbers were collected from the sample of members and shared anonymously with the LDM team, who secured the necessary Information Governance approvals from the NHS.

### 2.3. Stage 3: Data Linkage

The NHS numbers were pseudonymised by the NHS Data Service for Commissioners Regional Office (DSCRO) and returned to the LDM team to locate within the LDN dataset. 

### 2.4. Stage 4: Data Analysis

Variables related to demographics and health status were extracted from the LDM (see Table 2) and categorised to enable descriptive analysis. These variables were selected to provide a generalised picture of an individual’s personal characteristics and health as recorded by the health and care system. 

Descriptive analysis was undertaken in two stages. First, frequencies were produced for each variable to understand the distribution at a population level and identify any anomalies in the data. No anomalies were found, and no data were excluded. Second, each variable was cross tabulated to compare the LNN cohort with the Leeds population aged 64 and over. The outcome of stage two is presented and discussed in Section 3. This approach enabled the research team to draw inferences about how LNN participation varied from the wider population of the same age. Descriptive demographic data were used to answer RQ1, and health status data were used to answer RQ2. RQ3 was addressed through a discursive interpretation of the findings.

## 3. Findings

This section presents the findings of the article in relation to research questions 1 and 2.

### 3.1. RQ1: What Are the Demographic Characteristics of LNN Members and How Do They Compare to the Wider Leeds Population Aged 64 and over?

Table 3 presents the demographic characteristics of the LNN member cohort in relation to age, gender, ethnicity and the Indices of Multiple Deprivation (IMD) of the lower super output area (LOSA) in which they live. For each variable, the LNN cohort is compared with the remainder of the Leeds population aged 64 and over. These data suggest the following about LNN members: They tend to be older: 70 percent of the cohort were aged 80 or older compared to only 26 percent of the Leeds population aged over 64;They are more likely to be female: 75 percent were female compared to 54 percent of the Leeds population aged over 64;LNNs cater to a higher proportion of people of White ethnic backgrounds: 99 percent were White compared to 91 percent of the Leeds population aged over 64 (although it should be noted that the selected LNN was based in a predominantly White British area of the city).

These data also show that this LNN caters to older people from a cross-section of economic backgrounds, including people in the poorest and wealthiest communities according to the Indices of Multiple Deprivation (IMD): 14 percent were from the poorest 30 percent of communities compared to 30 percent of the Leeds population aged over 64, whilst 55 percent were from the most prosperous 30 percent of communities compared to 31 percent of the Leeds population aged 64 and over.

### 3.2. RQ2: What Are the Health Characteristics of LNN Members According to Key Population Health Measures and How Do They Compare to the Wider Leeds Population Aged 65 and Over?

The following sections present data on the health characteristics of LNN members using key variables in the LDM dataset. These measures are used within Leeds to aid planning and decision making in relation to health and social care integration and Population Health Management.

#### 3.2.1. Electronic Frailty Index Score

The Electronic Frailty Index (EFI) uses the information within the electronic primary healthcare record to identify populations of people aged 65 and over who may be living with varying degrees of frailty [19,20,21]. When applied to a local population it provides the opportunity to predict who may be at greatest risk of adverse outcomes in primary care as a result of their underlying vulnerability. It records a ‘cumulative deficit’ model to measure frailty on the basis of the accumulation of a range of deficits. These deficits include clinical signs (e.g., tremor), symptoms (e.g., vision problems), diseases, disabilities and abnormal test values [19].

EFI is presented as a score between 0 and 1 based on the number of deficits present out of a possible total of 36, with the higher scores indicating the increasing possibility of a person living with frailty and hence vulnerability to adverse outcomes. The scores are then categorised into levels of severity, with 0–0.12 meaning ‘fit’; >0.12–0.24 meaning ‘mild frailty’; >0.24–0.36 meaning ‘moderate frailty’; and above 0.36 meaning ‘severely frail’. Figure 1 presents the range of EFI scores in the LNN cohort compared with the Leeds 64+ population. 

This shows that levels of frailty amongst the LNN cohort were broadly similar to the wider population. In total, 85 percent were ‘fit’ or had ‘mild frailty’ compared to 83 percent of the wider population, 8 percent were ‘moderately frail’ compared to 12 percent of the wider population, and 7 percent were ‘severely frail’ compared to 6 percent of the wider population.

#### 3.2.2. Number of Long-Term Health Conditions (LTCs)

The LDM records the total number of diagnosed LTCs for each individual. Figure 2 presents the number of LTCs amongst the LNN cohort compared to the Leeds 64+ population. This shows that members of the pilot LNN tended to have more LTCs than the wider population, but that the difference was not that large. Overall, 8 percent of the LNN cohort had no LTCs compared to 16 percent of the wider population, 27 percent had one or two LTCs compared to 22 percent of the wider population, 19 percent had three or four LTCs compared to 21 percent of the wider population, and 19 percent had more than five LTCs compared to 17 percent of the wider population.

### 3.3. Public Health Management Cohort

The Public Health Management (PHM) process [21,22] segments the Leeds population according to levels of assessed health functioning and the stage of healthy ageing to aid the targeting of health and social care services. Four categories are used: ‘healthy’, to indicate a high level of functioning; ‘LTC’, to indicate the prevalence of at least one health condition that may affect frailty in the longer term; ‘frailty’, to indicate decreased functioning that is considered to be of concern with regard to functioning; and ‘end of life’, to indicate cases where functioning has reduced to the extent that an individual is included on the palliative care register. These categories are recorded in the LDM. Figure 3 presents the distribution of these PHM categories across the LNN cohort compared to the Leeds 64+ population. This shows that the largest proportion of LNN members (76 percent) fall into the LTC category, and that LNNs support a higher proportion of older people in this category than is present in the wider population (66 percent). It also shows that there are a smaller but significant proportion of LNN members in the frailty category (14 percent), but that this is slightly lower than the wider population (17 percent). 

## 4. Discussion: Implications for Role of Community Organisations Addressing Healthy Ageing as Part of Integrated Place-Based Care (RQ3)

This study aimed to understand the contribution of community organisations to healthy ageing and integrated place-based care through the case of the LNNs in Leeds, UK. The data about the demographic and health status of LNN members demonstrate that a significant volume of their work is targeted at older people at the later end of the age spectrum (i.e., aged 80 or older), many of whom are experiencing the onset of LTCs and/or mild frailty. This supports the central premise of integrated place-based care that many people’s health and care needs can be met in non-clinical community settings through services and activities provided by community organisations. By providing services that meet older people’s basic needs, offering opportunities for them to make decisions about their care, enabling mobility, and providing physical spaces in which they can develop relationships with peers and contribute to society, these organisations actively support older people to develop and maintain the core functional abilities that will promote wellbeing in older age [2]. 

Community organisations contribute to age friendly environments in which older people can age healthily in place, reducing barriers to participation and mitigating declining intrinsic capacity. This argument is supported by qualitative evidence from the wider study on LNNs [17,18] which suggests that community organisations can contribute to healthy ageing and integrated place-based care by delaying the onset in severity and complexity of health conditions, thus preventing or delaying transitions to more intensive forms of care as people age, and mitigating impact of those transitions when they do occur.

Although, the majority of LNN members do not present with acute health conditions or care needs, the LDM data suggest that LNNs do support a small number of members with more complex needs; seven percent were recorded as having severe frailty 38 percent had multiple LTCs (three or more). For these older people, engagement with community organisations is likely to be supplementary to the clinical care they receive. Given that growing the number of older people with moderate and more complex frailty who receive community-based support is a key aim of integrated place-based care [11,12] this evidence suggests that community organisations do have the capacity and capabilities necessary to support these individuals if they are appropriately resourced to do so.

The findings of this study also have practical implications. They demonstrate the value of joined-up data across local authorities and the NHS to support the integration of health and social care services. The LDM enabled beneficiaries of community organisations (LNN members) to be systematically identified within health and social care datasets and facilitated analysis of their demographic and health characteristics according to widely used measures. To our knowledge, this is the first time that integrated care datasets have been used in this way as community organisations and their beneficiaries are not routinely identified in these types of data currently. If they can be identified on a routine basis, it would enable a more consistent and comprehensive approach to research, insight and evaluation and may support the development of new and innovative services to tackle the problems facing communities, maximising the opportunities for health improvement [14]. The lessons from this study suggest that this will only be possible if community organisations such as the LNNs are supported to collect NHS numbers on a routine basis (i.e., upon first joining a service or activity) along with the appropriate consent. There also needs to be a commitment from Integrated Care System partners (i.e., local authorities and NHS) to pseudonymise and link these data with systems such as the LDM on a regular basis. Whilst this might not be feasible for all community organisations it ought to be prioritised in examples similar to the LNNs, where they are being commissioned to deliver key components of integrated place-based care.

## 5. Conclusions

This article has demonstrated the role community organisations can play to support healthy ageing and the place-based integration of health and social care. The findings suggest that community organisations, such as the LNNs, can support these ambitions in two main ways: first, by providing preventative support for older people with mild to moderate health and care needs, which may delay transitions to more intensive forms of care as people age; and second, by supporting older people with more severe needs, such as late cognitive or physical frailty, as a supplement to acute clinical care to enable to them to live in the community for longer. In light of these findings, healthcare systems in the UK, and globally, should consider how they might ensure the wider involvement of community organisations in integrated place-base care, for example, by providing more consistent funding for their work.

We also note some limitations to our study, which prevent generalisation beyond our research location of Leeds, UK. First, the LNN model is unique to Leeds. No other area of the UK (or internationally, as far as we are aware) has a similarly scaled model in which community organisations receive core public sector funding to support integrated place-based care. Second, due to resource constraints, we were only able to collect data on members from one LNN (of 37) in a predominantly White population. Each neighbourhood of Leeds has different social, economic and demographic characteristics, meaning that the needs and circumstances of members will vary from one network to another. Third, it would have been beneficial to include a wider range of LDM variables, such as those relating to mental health. This should be considered for future studies. Finally, our study was observational, covering one point in time (2021–22) which means it may be subject to bias and confounding factors. Future studies should track changes in LDM variables longitudinally to understand the extent to which participation in an LNN mitigates deterioration in various health measures relative to the wider population. This was beyond the scope of our study but ought to be a priority for future research.

## Figures and Tables

**Figure 1 healthcare-11-02827-f001:**
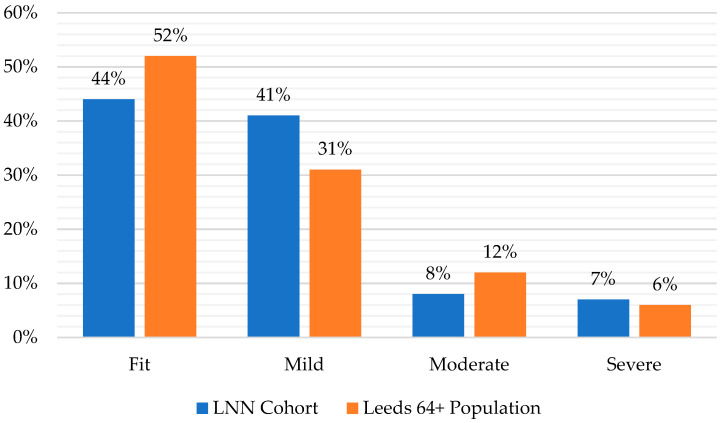
EFI score for the LNN cohort compared with the Leeds 64+ population. Source: Leeds Data Model, June 2022. Base: LNN Cohort, 148; Leeds 64+ Population, 143,418.

**Figure 2 healthcare-11-02827-f002:**
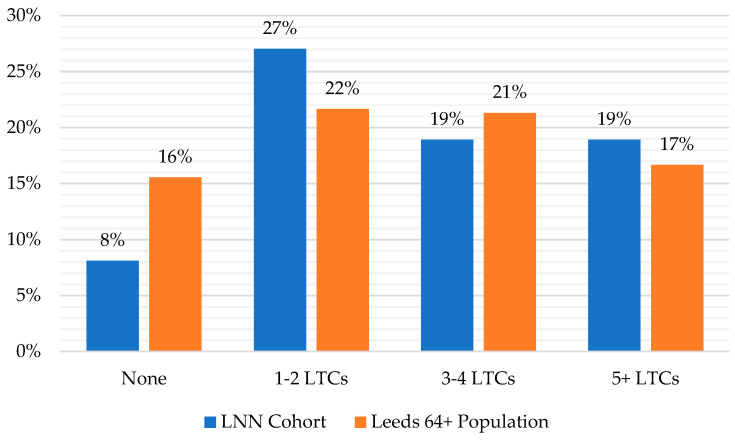
Number of LTCs amongst the LNN cohort compared with the Leeds 64+ population. Source: Leeds Data Model, June 2022. Base: LNN Cohort, 148; Leeds 64+ Population, 143,418.

**Figure 3 healthcare-11-02827-f003:**
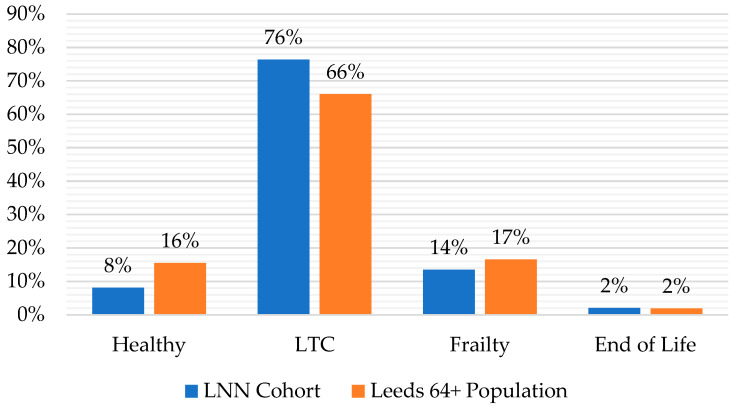
PHM categories of the LNN cohort compared with the Leeds 64+ population. Source: Leeds Data Model, June 2022. Base: LNN Cohort, 148; Leeds 64+ Population, 143,418.

**Table 1 healthcare-11-02827-t001:** An overview of LNN services and activities by category.

Category	Subcategories	Link to WHO Functional Abilities
Opportunities for social connection and interaction	Shared hobby or interest groups	Build and maintain relationships, Be Mobile
	Lunch club or café	
	Door-step visits during the COVID-19 pandemic	
	Games	
	One-off events	
	Day trips and holidays	
Support to engage in physical activity or exercise	Sport or recreation activities	Build and maintain relationships, Be Mobile
	Fitness classes and activities	
Learning and development opportunities	Hobby or interest	Learn, grow and make decisions
	Fitness	
	IT/Digital	
	Religion/faith	
Befriending	Telephone	Build and maintain relationships
	Face-to-face	
Food and nutrition	Shop	Meet basic needs
	Lunch clubs	
	Medicines collection	
	Shopping and food delivery	
	Meals on wheels	
Transport	Car pick-ups	Meet basic needs, Build and maintain relationships, Be Mobile
	Minibuses	
Frailty and long-term conditions clinics	Strength and balance	Build and maintain relationships, Be Mobile
	Leg clinics	
	Memory loss	
Information, advice and guidance	Newsletters	Meet basic needs
	Websites	
	One-to-one sessions	
	Referral to other agencies	
	Signposting to other agencies	
	Housing	
	Benefits	
Home improvement and adaptation	Handyman services	Meet basic needs, Be mobile
	Gardening	
	Fire safety checks	
Volunteering opportunities across a range of services		Contribute to society

Reproduced from Dayson et al., 2022 [17].

**Table 2 healthcare-11-02827-t002:** Overview of LDM variables.

Demographics	Health Status
Age	Frailty score (EFI)
Gender	No. of long-term health conditions
Ethnicity	Population Health Management (PHM) cohort
IMD decile	

**Table 3 healthcare-11-02827-t003:** (a) Demographic characteristics: age. (b) Demographic characteristics: gender. (c) Demographic characteristics: ethnicity. (d) Demographic characteristics: Indices of Multiple Deprivation.

**(a)**				
**Age Category**	**LNN Cohort**		**Leeds 64+ Population**	
	**Count**	**Percent**	**Count**	**Percent**
60 to 64	1	0.7	8274	5.8
65 to 69	5	3.4	35,862	25.0
70 to 74	23	15.5	34,766	24.2
75 to 79	16	10.8	26,699	18.6
80 to 84	38	25.7	18,163	12.7
85 to 89	34	23.0	12,367	8.6
90 to 94	22	14.9	5570	3.9
95+	9	6.1	1717	1.2
**Total**	**148**		**143,418**	
**(b)**				
**Gender**	**LNN Cohort**		**Leeds 64+ Population**	
	**Count**	**Percent**	**Count**	**Percent**
Female	111	75.0	77,140	53.8
Male	37	25.0	66,278	46.2
**Total**	**148**		**143,418**	
**(c)**				
**Ethnicity**	**LNN Cohort**		**Leeds 64+ Population**	
	**Count**	**Percent**	**Count**	**Percent**
White Background	147	99.3	130,668	91.1
Asian Background	1	0.7	4838	3.4
Unknown	0	0.0	4130	2.9
Black Background	0	0.0	1836	1.3
Chinese or Other Background	0	0.0	1185	0.8
Mixed Background	0	0.0	761	0.5
**Total**	**148**		**143,418**	
**(d)**				
**IMD Decile**	**LNN Cohort**		**Leeds 64+ Population**	
	**Count**	**Percent**	**Count**	**Percent**
1	9	6.1	22,212	15.5
2	11	7.4	10,087	7.0
3	0	0.0	10,519	7.3
4	0	0.0	5058	3.5
5	20	13.5	12,205	8.5
6	2	1.4	12,710	8.9
7	22	14.9	18,564	12.9
8	6	4.1	14,082	9.8
9	28	18.9	15,955	11.1
10	48	32.4	13,753	9.6
-	2	1.4	8273	5.8
**Total**	**148**		**143,418**	

## Data Availability

Additional data for this study are not publicly available due to the conditions under which it was shared by Leeds Office of Data Analytics (part of Leeds Health and Care Partnership and NHS West Yorkshire Integrated Care Board). Please contact the authors for further information.

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
