# Peer review of "Understanding the Contribution of Community Organisations to Healthy Ageing and Integrated Place-Based Care: Evidence from Integrated Care Data"

_healthcare, 2023, doi:10.3390/healthcare11212827_

Round 1

Reviewer 1 Report

I appreciatte the opportunity to review this work precisely in this importat moment for the healthy aging. Is important not only promote health in adolescene or adulthood, but not forget older people. So congratulations! So, my comments are minimal. I would like to know why your do not include information about their mental health, maybe one indicator of mental health knowing their importance. If you don´t measure some indicator maybe you could mention this factor in the limitation or maybe prospect of the future, and how to promote these models of health care to other communities with fewer opportunities. I would like to know the results in the future. 

Author Response

Thank you for you comments. We agree mental health is an important consideration but  these measures were not included the Leeds Data Model extract we analysed. We have noted this as a limitation in the revised manuscript.

Reviewer 2 Report

Title: Understanding the Contribution of Community Organizations to Healthy Ageing and Integrated Place-Based Care: Evidence from Integrated Care Data.

Reviewer Comments: Purpose of this study to study the contribution of community-based support through LNNs. 148 LNN beneficiaries participated, and population aged over 64 included in this study. Leeds data model of was used in this study. Demographic characteristics, EFI, LTCs, and public health management cohort categorizations were measured. LNN’s are focused on older people who are fit (44%) or experiencing the onset of LTCs (27%) 19 and/or mild frailty (41%). Community organizations are good enough to support the ambitions of integrated care by providing support for older people with mild to moderate health problems. They also have the capacity to support older people with more serious needs if they have enough resources.

Strengths:

1.     Author did a good job in addressing vast range of stress factors.

2.     Sample size in the study group.

3.     Using Integrated care datasets in community organization studies.

4.     Most of concerns were mentioned as limitations in this article.

Weaknesses:

1.     There are few minor spellings mistakes Ex: in Line 18.

2.     Authors mentioned that community organizations have the capacity and capability to support older people with more severe needs as a supplement to acute clinical care. Authors need to elaborate severe needs. What kind of severe needs. One or two examples would be helpful.

3.     Since this study is observational study there is a lower standard of evidence compared to experimental studies. These types of studies are more prone to bias and confounding.

4.     This study has been monitored exclusively in white population. It would have been more informational, had it been conducted in mixed population.

5.     In this study, did they conduct any patient satisfaction survey? If yes what questions were asked?

6.     How long these LNNs provide community-based service for elderly people? Is it long term or only specific amount of time.

7.     What are the limitations to apply this concept at other places or countries?

Author Response

Thank you for you comments. We have revised the manuscript as follows.

1. We have reviewed for spelling mistakes etc

2. We have included the examples of 'late cognitive or physical frailty' in the conclusion

3-4. We have now explicitly acknowledged this in the limitations section.

5. There was not a patient satisfaction survey but some qualitative research with patients was undertaken. These findings are reported in other outputs referenced in our article.

6. For most members long-term support is provided, from periods ranging from 1 year to more than 10 years in some instances. We have added a sentence to this effect.

7. This is a very good question but not one we feel able to answer at this stage as we have not studied the integrated care context in other countries. However, this is a focus of our current research.

Reviewer 3 Report

Dear Authors,

We recently had the opportunity to read your manuscript titled “Understanding the Contribution of Community Organisations to Healthy Ageing and Integrated Place-Based Care: Evidence from Integrated Care Data”, and we wanted to reach out to you to express our comments about your work.

This observational study explored the contribution of community organizations to healthy aging and integrated care in Leeds, UK. Using the Leeds Data Model, the health characteristics of 148 members of the Leeds Neighbourhood Networks were analyzed and compared to the wider Leeds population aged 64+. Findings suggest these organizations support older adults with mild-moderate needs, delaying transitions to more intensive care. They also have capacity to support those with more complex needs. Overall, community organizations are well-positioned to enable healthy aging in place through social and preventative services.

Nevertheless, here are some possible comments outlining areas that could improve the quality and readability of the manuscript:

Introduction:

1.      Research aims could be more clearly and succinctly stated upfront.

2.     Prior research reviewed seems sufficient but could cite any key quantitative studies.

Methods:

3.     Participant recruitment/sampling methods should be described.

4.     Data analysis methods could be more clearly explained. Please, expand it.

5.     Limitations of observational design should be acknowledged.

Findings/Results:

6.     Findings align with aims though objectives could be clarified.

7.     Appears comprehensive but methods should state if any data is excluded.

Discussion:

8.     Summarizes key findings but needs to directly tie back to aims stated in the introduction.

9.     Does not acknowledge observational design limitations.

Conclusions:

10.  Research aims are partially addressed but could be tied together more clearly. Please, detail this part.

11.  The significance and implications are described in a somewhat lengthy way. Please, correct it.

12.  The main takeaways could be distilled and strengthened.

Regarding the grammar and the use of English language, the manuscript is well-written overall, with clear and concise language. A few minor edits could improve clarity:

·      Some verbosity in the introduction could be tightened up with more direct phrasing.

·      Acronyms like LNN and LDM are defined but should be written out in full the first time mentioned.

·      There are some lengthy sentences that could be shortened for readability.

·      The conclusion effectively summarizes the main points but could be made more succinct.

Once again, thank you very much for your work. We´ll be waiting for your answers about our comments.

Kindest regards,

Regarding the grammar and the use of English language, the manuscript is well-written overall, with clear and concise language. A few minor edits could improve clarity:

·      Some verbosity in the introduction could be tightened up with more direct phrasing.

·      Acronyms like LNN and LDM are defined but should be written out in full the first time mentioned.

·      There are some lengthy sentences that could be shortened for readability.

·      The conclusion effectively summarizes the main points but could be made more succinct.

Author Response

Thank you for you comments. Please find our response below.

  1. We now included the following explicit aim prior to the research questions: "This article aims to understand the contribution that community organisations to healthy ageing and integrated place-based care".
  2. We have added an opening paragraph to include some key statistics about the global ageing population.
  3. We have added more detail about sampling in the methods section.
  4. We have rewritten our approach to data analysis to provide more clarity about the process.
  5. We have elaborated on the limitations of the study in the final paragraph.
  6. We have made it clearer which data are answering which research question
  7. No data were excluded. This is no stated in the analysis section.
  8. The discussion section is intended to answer RQ3. We have now made this clear and restated the aims of the study.
  9. See response to 5, above.
  10. We have made a number of revisions to the conclusion to address this point and points 11-12.

In addition, we have noted the comments around grammar and language and sought to address these through a thorough proof read.

Reviewer 4 Report

Dear Authors,

It is interesting topic. 

The aim of this article is described in general way (lines 44-48) as filling the gap "in the context of the UK where the integration of placed-based primary and secondary health care (which is the remit of the universal National Health Service - NHS) with social care (which is the preserve of local government) is a long-term strategic policy goal."

There are also 4 research questions but in fact there is no clearly specified goal of this research apart from that to fill the gap in the context of ... .

Also, you should try to show practical implication of this research in general term. Of course you have underlined that it is specific but still some practical implication should be mentioned and showed. What is the value of such research in the wider context (range).

Author Response

Thank you for this comments.

In the Introduction, we have now more clearly spelled out the aim of our study prior to introducing the research questions.

In the Discussion, we have been more specific about the practical implications of our study. In the Conclusion we have also included recommendation for further funding for community organisations providing such care.

Round 2

Reviewer 4 Report

All comments have been taken into account. It improved the clarity of article.